# SCADA-Based Heliostat Control System with a Fuzzy Logic Controller for the Heliostat Orientation

**Eugenio Salgado-Plasencia, Roberto V. Carrillo-Serrano, Edgar A. Rivas-Araiza and Manuel Toledano-Ayala ***

División de Investigación y Posgrado, Facultad de Ingeniería, Universidad Autónoma de Querétaro, Cerro de las Campanas s/n, Santiago de Querétaro C.P. 76010, Querétaro, Mexico

\* Correspondence: toledano@uaq.mx; Tel.: +52-442-144-5820

**Abstract:** In central receiver systems, there are local controls that modify the position of the heliostats, where the trend is to increase the intelligence of the local controls in order to give them greater autonomy from the central control. This document describes the design and construction of a SCADA (Supervisory Control and Data Acquisition)-based heliostat control system (HCS) with a fuzzy logic controller (FLC) for the orientation control. The HCS includes a supervisory unit with a graphical user interface, a wireless communication network, and a stand-alone remote terminal unit (RTU) implemented on a low-cost microcontroller (MCU). The MCU uses a solar position algorithm with a maximal error of 0.0027° in order to compute the position of the sun and the desired angles of the heliostat, according to a control command sent by the supervisory unit. Afterwards, the FLC orients the heliostat to the desired position. The results show that the RTU can perform all the tasks and calculations for the orientation control by using only one low-cost microcontroller with a mean squared error less than 0.1°. Besides, the FLC orients the heliostat by using the same controller parameters in both axes. Therefore, it is not necessary to tune the controller parameters, as in the traditional PID (Proportional-Integral-Derivative) controllers. The system can be adapted in order to control other two-axis solar-tracking systems.

**Keywords:** SCADA system; sun tracking; heliostat; orientation control; fuzzy logic controller; solar energy

## 1. Introduction

The use of renewable energy resources has grown as a result of the concerns about $CO_2$ emissions, dependence on fossil fuels, as well as climate change. Solar energy is a clean and practically infinite renewable resource, which could satisfy the global energy demand with only a small fraction of the solar radiation that reaches the surface of the Earth. Therefore, the development of new efficient ways to use solar energy at accessible costs is both a great scientific and technological opportunity [1].

In high-concentration solar plants, it is necessary to have control systems that help to reduce losses in the concentration of solar radiation and to avoid damage to the operators and the facilities. In order to collect solar energy, it is necessary to know the position of the sun throughout the day. This can be achieved by using a solar tracker, which combines a mechanical structure with electronic feedback devices. Solar trackers are divided into single-axis or two-axis trackers. In single-axis solar trackers, the tracking system drives the rotation axis until the normal component of the mechanical structure and the solar vector are coplanar, whereas the double-axis solar trackers follow the sun in the horizontal and the vertical plane [2]. The solar tracking systems are also divided into three classifications: the sensor driver systems (SDS), which determine the position of the sun using solar

sensors, the microprocessor driver systems (MDS), which use an algorithm to determine the position of the sun using a microcontroller, and the combination of the first two methods [3].

Dual-axis SDS sun trackers have been designed in order to drive photovoltaic (PV) systems. Seme et al. [4] developed a dual-axis tracking system without using a microcontroller, producing more than 27% electric energy than a fixed system. Ahmed [5] and Rao et al. [6] implemented a dual-axis sun tracker on an 8051 MCU, using light dependent resistors (LDR) to sense the position of the sun and two stepper motors to orient a PV panel. Mishra et al. [7] designed a solar tracking system based on Arduino UNO with a Bluetooth radio module to get a real-time measurement of the output voltage of the solar panel on a mobile app. Makhija et al. [8] used an ATMega328 microcontroller, four LDRs, and two servo motors to build a sun tracker on a PV, increasing the output voltage 37% against a fixed PV. Flores-Hernández et al. [9] developed a dual-axis solar tracker using a robotic sensor, allowing to reduce the energy consumption of the tracking system as well as adding multiple tracking systems using the same sensor.

Other works implemented MDS sun trackers. Chabuk et al. [10] applied a dual-axis solar tracker based on an MCU, using a real-time clock (RTC) to obtain the time of the day and to determine the angle of movement of two stepper motors. Yang et al. [11] developed an open-loop sun tracker controlled by a microprocessor AT89S52, using a GPS and an RTC to determine the geographical position coordinates (latitude and longitude) and the current date and time to obtain the sun position. Sidek et al. [12] built a master-slave dual-axis solar tracking system using two slave microcontrollers to control the position of both axes, which are governed by a master microcontroller, obtaining an accuracy of 0.5° in the sun tracker with a maximum energy gain of 26.9% better than the fixed-tilted PV panel. Kumar.N and Subramaniam [13] implemented an MDS sun tracker to orient a solar dish concentrator, calculating the sun position using an RTC and a solar position equation, thus obtaining 75% more average thermal energy compared to a fixed solar tracking system.

In central receiver systems, there are two-axis solar tracking reflectors called heliostats that concentrate the sunlight on a receiver located on top of a tower. Every heliostat uses a local control that guides the heliostat into the desired position by using two electric motors connected to reduction gears. A control algorithm modifies the angular position of the motors, allowing the mechanical structure to remain a stable position despite changes in the environmental conditions [1]. Besides, a unique central control is used to communicate with the local controls through a wireless communication system. However, due to the limited bandwidth in wireless communication, it is difficult to control the orientation of each heliostat from the central control with a high update rate, which increases the possibility of errors. Therefore, every local control must calculate the sun position and the aim point in order to move the heliostat to the desired position, reducing the network traffic between the central and the local control, and increasing its autonomy [14]. A SCADA system integrates the information received from the local controllers (also called remote terminal units), by using a supervisory unit (SU) embedded into a computer, in order to analyze the data, supervise the process, and make decisions by using a human-machine interface [1].

Some heliostat control systems have been presented in the literature. Loudadi and El Omari [15] developed a heliostat control by using Java language on a computer, which calculates the angles of the position of the sun and the heliostat, sending them via Bluetooth to a microcontroller-based card, in order to control a 0.16 m$^2$ heliostat by using two low-power DC motors. The heliostat sends its actual position to the computer, which compares it with the calculated values and corrects the position of the heliostat if there is a difference between the values. Pışırır and Bıngöl [16] developed a heliostat control in an embedded system using an industrial computer model ARK 1388V, which performs all the tasks of the heliostat. The control system orients a 1 m$^2$ heliostat by using two stepper motors, a programmable controller, and a driver circuit. The system moves the heliostat every four minutes, by using the previous and new position of the heliostat in order to calculate the step number of the motors. Gross et al. [17] developed a control system with a graphical user interface (GUI) to monitor a

field of heliostats, allowing to control multiple receivers and types of heliostats in a single application. The system can be updated by adding new heliostats in the field and more than one receiver.

Finally, an advanced control strategy, such as a fuzzy logic controller, is an excellent alternative to deal with changing dynamics and non-linearities present in solar plants that traditional controllers cannot handle [1]. Fuzzy logic controllers have been applied in order to control the orientation of a heliostat. Zeghoudi and Chermitti [18] applied a speed control of a DC motor with an FLC for the orientation of a heliostat. Two FLC configurations were compared with a neural controller in Matlab software. The first configuration (FLC1) uses two inputs (error and derivate of the error), one output, and a rule base of ten rules, whereas second configuration (FLC2) uses only one input (error), one output, and a rule base three rules. The results demonstrated a better output response for the FLC compared with the neural controller, which presents some fluctuations that can be harmful to the DC motors. Zeghoudi et al. [19] applied an adaptive fuzzy-proportional-integral (F-PI) controller in order to orient a heliostat by controlling a DC motor in Matlab software. The F-PI controller was compared with two FLC configurations (FLC1 and FLC2) and a conventional Proportional-Integral (PI) controller. The F-PI controller modifies the controller gains of a PI controller, by using two inputs (error and derivate of the error), two outputs (proportional gain and integral gain) and a rule base of thirty-five elements for each output. The results showed that the F-PI controller gives the best results compared to other methods. However, these works have only been presented in simulations.

Taking into account the developed works, this paper describes the design and construction of a heliostat control system based on a SCADA system with a stand-alone remote terminal unit and an embedded fuzzy logic controller on a low-cost microcontroller for the orientation control.

## 2. Materials and Methods

### 2.1. Heliostat Control System

The HCS controls the heliostat in order to reflect the solar radiation on the desired target. It is a mechatronic system, which consists of mechanical, electronic, and communication units. The block diagram of the HCS is shown in Figure 1. The HCS proposed consists of a supervisory unit with a GUI, a mobile app, a wireless communication network with radio frequency (RF) XBee modules and a Bluetooth radio module, and an MCU-based RTU connected to the mechanical structure of the heliostat.

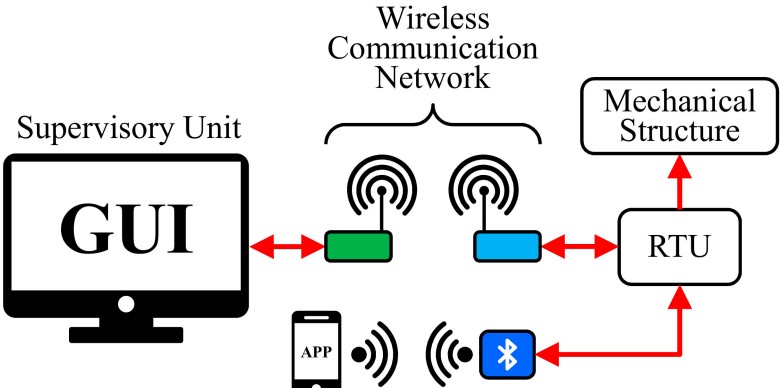

**Figure 1.** Block diagram of the heliostat control system.

### 2.2. Mechanical Structure

There are common features in the mechanical structure of the heliostats, such as facets, a support frame, and a driver mechanism [1]. The facets are mirrors that reflect the solar radiation to the desired target. The facets are attached to a support frame by anchor points that hold them in the desired position. Finally, the support frame is connected to the heliostat axes, which are driven by individual reduction mechanisms.

The design and parameters of the heliostat are shown in Figure 2 and Table 1, respectively. It is a heliostat with an azimuth-elevation mechanism [2] with a range of motion of 360 degrees on each axis. There are 16 facets attached to the support frame. In addition, the heliostat has a gap that allows directing the reflecting side of the mirrors to the ground. The heliostat is oriented by two DC motors, which are connected to the axes of the heliostat by worm drive mechanisms. There are also two rotary encoders connected to the axes of the heliostat for the angular position feedback.

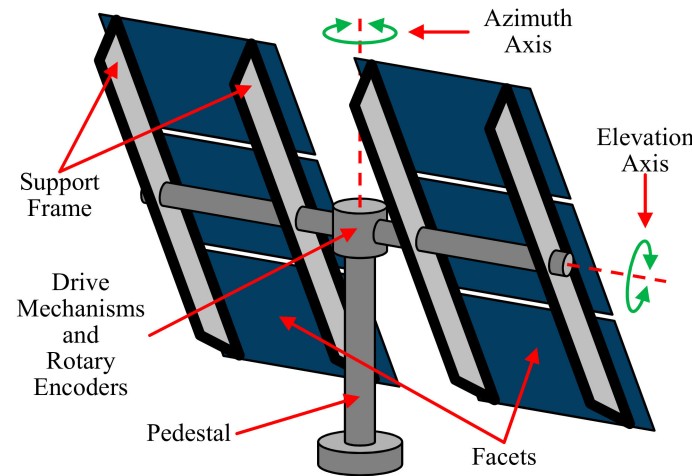

**Figure 2.** Heliostat design.

**Table 1.** Parameters of the heliostat.

| Parameter | Value | Unit |
|---|---|---|
| Pedestal height | 2.85 | m |
| Elevation axis length | 4.43 | m |
| Number of facets | 16 | - |
| Mirror face size | $1.2 \times 1.2$ | m |
| Total reflective area | 23 | m$^2$ |
| Gap between support frames | 0.70 | m |
| Total height | 5.24 | m |

### 2.3. Supervisory Unit

The SU supervises and modifies the operation status (OS) of the RTU, in order to orient the heliostat to the desired position. The OS is defined as follows:

- Stop. The heliostat stops and keeps its position.
- Initial Position. The heliostat moves to the resting position and keeps its position.
- Maintenance Position. The heliostat moves to the maintenance position and keeps its position.
- Safe Position. The heliostat directs the elevation axis parallel to the ground to reduce the load of the wind over the mechanical structure.
- Automatic Position. The heliostat reflects solar radiation on the target.
- Standby position. The heliostat reflects the solar radiation above the target to reduce the time of orientation in case of the OS changes to the automatic position.
- Manual Control. The heliostat is controlled by the operator.

#### 2.3.1. Graphical User Interface

The SU monitors and controls the OS of the RTU using a GUI implemented on a personal computer. The GUI runs on the Ubuntu operating system, and it was developed using the Python programming

language and the graphical user interface builder Glade Interface Designer. Figure 3 shows the algorithm of the GUI. At the start of the program, the modules of Python, the Glade file, and the data from the configuration files are loaded. The configuration files contain the constant values of the heliostat and the target (geographical position, distance to the target, heliostat height, and target height), the MAC address of the XBee RF modules, and the local weather record. After the referred sequence, the variables are initialized, and the action of the widgets are defined (buttons, labels, and images). Afterwards, a user ID and a password are required to access the main program.

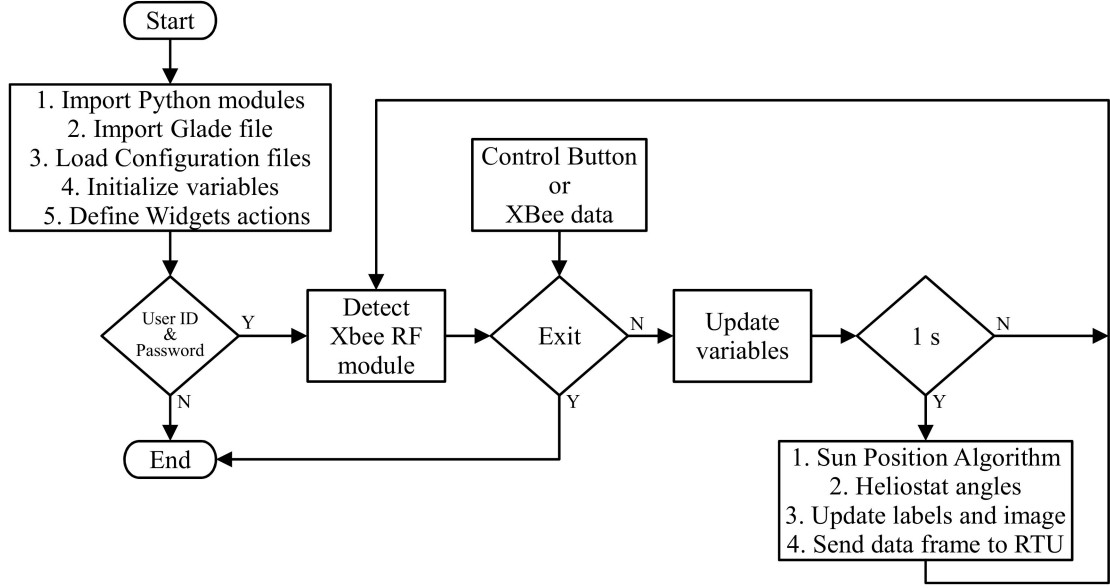

**Figure 3.** Algorithm of the graphical user interface.

Later, a 50 ms timer refreshes the main loop of the algorithm. The first stage of the loop is to detect the coordinator XBee RF module of the wireless communication network in order to send and receive data from the RTU. The next stage is to update the variables with the data received from the RTU and the control buttons of the GUI, which modify the OS of the RTU. If the exit command is selected, the program ends. If the program is not finished, the algorithm verifies if a second has elapsed. If the change occurs, the program executes the solar position algorithm for the geographical position of the target and the heliostat, calculates the heliostat angles in order to compare them with the data received from the RTU, updates the labels of the GUI, updates the image of the OS of the RTU, and sends a data frame with the desired OS to the RTU. Otherwise, the timer refreshes the main loop. The GUI also has an emergency stop button, which stops the heliostat when the operator detects a failure. When the emergency stop button is pressed, the heliostat remains in stop position until the button is released.

### 2.3.2. Mobile App

The heliostat can also be controlled via Bluetooth using a mobile app. The app modifies the OS of the RTU using the algorithm showed in Figure 4. At the start of the program, the app waits for the Bluetooth connection between the app and the RTU. When the Bluetooth connection is established, the app reads the data received and updates the variables and labels to show the status of the RTU on the mobile device. If a control button is pressed, the app sends a data frame to modify the OS of the RTU or corrects the data of the RTC. If the Bluetooth connection is not established and the exit command is selected, the program ends.

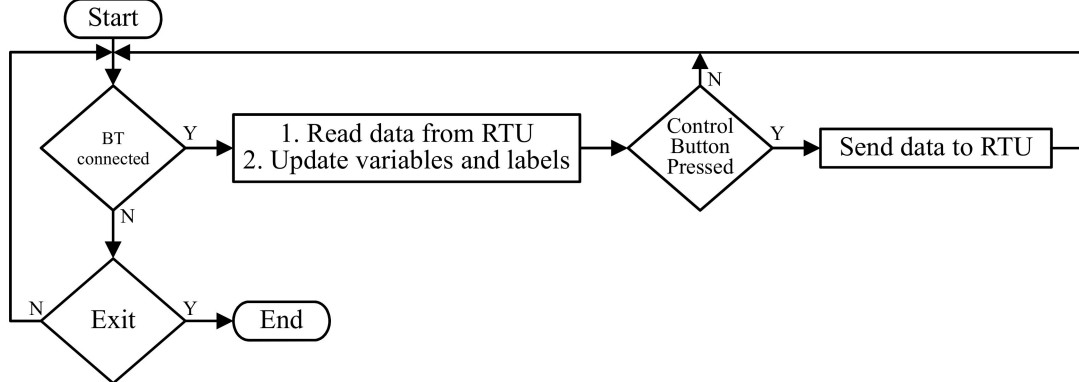

**Figure 4.** Algorithm of the Mobile App.

### 2.4. Remote Terminal Unit

The orientation of the heliostat is controlled by an RTU based on an MCU. The block diagram of the RTU is shown in Figure 5. The RTU consists of an MCU, an RTC, an XBee RF module, a Bluetooth radio module, an analog thumb joystick, a 20 × 4 liquid crystal display (LCD), two motor driver circuits for the two DC motors, two rotary encoders, and a programming port ICSP (In-Circuit Serial Programming).

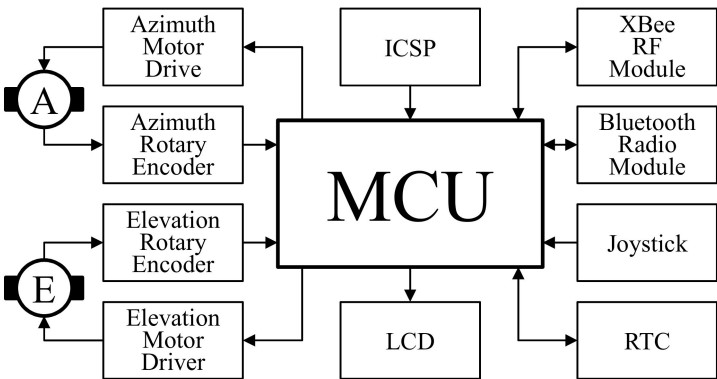

**Figure 5.** Block diagram of the remote terminal unit.

### 2.4.1. Microcontroller

The MCU is the central component of the RTU. The MCU used by the RTU is a dsPIC33EP256MU806, which is a 16-Bit digital signal controller designed for motor control. The MCU computes the position of the sun by using an astronomical solar position algorithm and calculates the angles of the heliostat to move it to the desired position according to OS received from the SU. Finally, the MCU controls the orientation of the heliostat using two pulse width modulation (PWM) signals that command the DC motors, comparing the desired position with the current position using the rotary encoders.

### 2.4.2. Real-Time Clock

An RTC is an essential component of the RTU because the solar position algorithm receives the time and date as input data. Therefore, any error in the RTC time impacts on the solar vector calculation. The RTU uses an RTC DS1307. It is an RTC with I2C (Inter-Integrated Circuit) serial interface, low-power consumption, leap year correction, and a Battery-Backed supply with automatic power-fail detection.

### 2.4.3. PWM Motor Driver

The current output of the MCU is too low to drive a high current DC motor, which could damage the output pins of the MCU that are not designed to drive high current devices. Therefore, a high

current driver interface is required to modify the speed and direction of the motor by using a control signal from the MCU and an external power supply.

The implemented motor driver for each DC motor is shown in Figure 6. It is an H-Bridge motor driver built with four bipolar junction transistors (Q1, Q2, Q3, and Q4). The H-Bridge modifies the direction of rotation of the DC motor by changing the direction of the current that flows through the motor by controlling transistors Q5 and Q6 as switches, whereas the speed is modified by two PWM signals for the two directions of rotation. Transistor Q5 allows current flow through transistors Q1 and Q4 for the clockwise direction of rotation, whereas transistor Q6 allows current flow through transistors Q2 and Q3 for the counterclockwise direction of rotation. However, activating both directions of rotation at the same time produces a short-circuit that could damage the components of the H-Bridge. Therefore, there must be a delay (called dead time) between both PWM signals, in order to avoid a short-circuit. Finally, there are four diodes for back electromotive force protection when the motor stops.

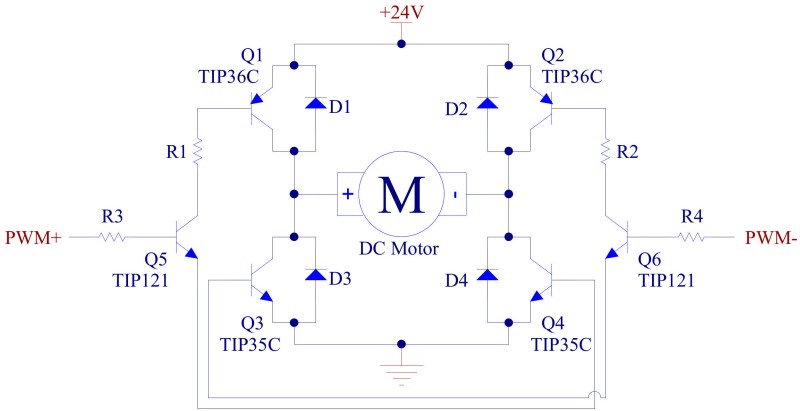

**Figure 6.** H-bridge driver circuit.

The components of the motor driver were selected in order to modify the speed and direction of a DC gear motors model ZYT6590-01. The parameters of the DC motor are presented in Table 2.

**Table 2.** Parameter of the DC motors.

| Parameter | Value | Unit |
| --- | --- | --- |
| Rated Voltage | 24 | V |
| No Load Current | 1 | A |
| Rated Current | 5 | A |
| Maximum Current | 25 | A |
| Rated Speed | 5 | rpm |

### 2.4.4. Rotary Encoders

The feedback signals of the RTU are generated by two absolute rotary encoders, which can maintain the angular position data when the power is removed from the system, recovering it when the power is applied back to the encoders. The system uses two single-turn absolute rotary encoders model CAS60RS12A10SGG with 12 bits of resolution (4096 pulses per revolution), synchronous serial interface (SSI), 5 V operating voltage, and binary code output.

### 2.4.5. RTU Algorithm

The control algorithm of the RTU is shown in Figure 7. At the start of the program, the geographical and field positions of the heliostat are read from a database. Afterwards, all the parameters of the different components of the MCU and the program variables are initialized. Four timers are used to execute interrupts in the main program at specific time intervals, as shown in Table 3. Besides, a

watchdog timer resets the MCU at any error in the main program. Finally, the functions of the external components of the MCU are initialized: two universal asynchronous receiver-transmitter (UART) modules for the serial communication with the XBee RF module and the Bluetooth radio module, an I2C module for the serial communication with the RTC, an ADC (Analog-to-Digital Converter) module for the manual control with the analog joystick, and the parallel communication with the 20 × 4 LCD.

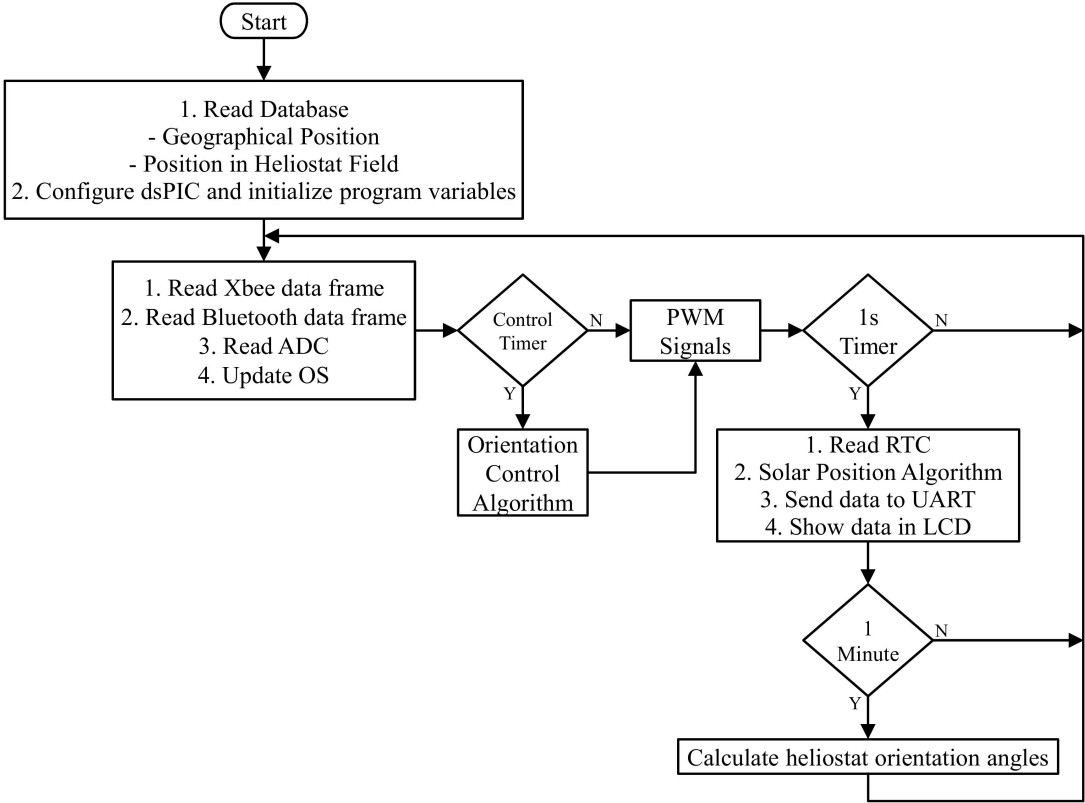

**Figure 7.** Control algorithm of the remote terminal unit.

**Table 3.** Timers of the remote terminal unit algorithm.

| Timer | Function | Frequency | Period |
|-------|----------|-----------|--------|
| 1 | Encoder clock signal | 200 kHz | 5 μs |
| 2 | PWM period | 1 kHz | 1 ms |
| 3 | Controller sampling time | 100 Hz | 10 ms |
| 4 | 1-s timer | 2 Hz | 500 ms |

In the main loop of the program, the control data is received from the XBee RF module, the Bluetooth radio module, or the analog joystick. The manual control has a higher priority than the data received from the Bluetooth radio module, which has a higher priority than the data received from the XBee RF module. However, if an emergency stop is received from the SU, the OS only can be changed by the SU. If no data is received in 20 s, a flag restarts the MCU, in order to reset the communication modules at any error in the program.

The orientation control algorithm modifies the PWM signals (with a dead time of 1 ms) when the control timer (Timer 3) overflows, in order to orient the heliostat to the desired position, which is determined by the received OS. Afterwards, every second the program reads the date and time from the RTC module, calculates the position of the sun, sends the data to the SU, and shows the OS in the LCD. If one minute has elapsed, the program calculates the angles of the heliostat; this is because the sun changes its position in the sky by 0.25° every minute. Therefore, it is not necessary to change the angular position of the heliostat every second of the day.

### 2.5. Wireless Communication Network

The RTU is controlled by receiving a data frame from the GUI and the mobile app, using an XBee RF module and a Bluetooth radio module, as shown in Figure 8. This allows an operator to visualize and modify the OS of the RTU.

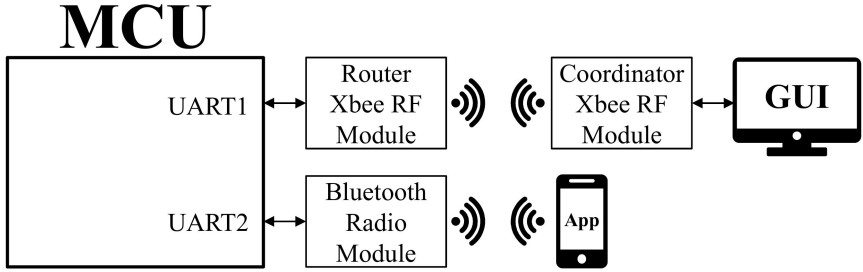

**Figure 8.** Block diagram of the wireless communication network.

### 2.5.1. XBee RF Module

Two XB24CZ7WIT-004 XBee RF modules are used for the communication between the SU and the RTU, one as coordinator and the other as a router. The XBee RF modules use the IEEE 802.15.4 networking protocol and operate within the ISM 2.4 GHz frequency band. Besides, the modules communicate with each other by using a full mesh protocol with a data rate of 250 Kbps and a maximum number of elements of 65,536.

The XBee RF modules communicate with each other by sending and receiving wireless messages. However, they send and receive data through a serial interface to the connected device by using the API (Application Programming Interface) operating mode. The router XBee RF module communicates with an MCU through a serial UART interface, whereas the SU communicates with the coordinator XBee RF module by using a serial module. The API operating mode requires that communication with the module be done through a structured interface, that is to say, that the data is transmitted via API frames [20]. The API frame structure consists of a start delimiter byte, two length bytes, the frame data, and a checksum byte for error detection. The API operating mode allows the RF modules to communicate with each other by using a point-to-point model (one sender and one receiver) or a broadcast model (one sender and multiple receivers), only by modifying the frame type data in the frame of data.

The data frame from the RTU to the SU includes the heliostat ID, the RTU OS, the Bluetooth connection status, the angles of the sun position, the current angles of the heliostat, and the date and the time of the RTC. For communication from the SU to the RTU, the frame modifies the OS of the RTU, corrects the RTC data, or restarts the MCU.

### 2.5.2. Bluetooth Radio Module

The Bluetooth module is the RN41XVC-I/RM, which is a Class 1 module, with a data rate of 240 Kbps and configuration via local UART and wireless RF. The Bluetooth radio module communicates with the MCU through a serial UART interface.

The data sent from the RTU to the App includes a start and a final character to distinguish the different values contained in the data frame. The data frame consists of the encoder 1 data, the encoder 2 data, the OS, the date and time of the RTC, and the resolution of the encoders. For communication from the mobile app to the RTU, the frame modifies the OS of the RTU, corrects the RTC data, or restarts the MCU.

### 2.6. Orientation Control

In order to guide the heliostat to the desired position, the MCU calculates the position of the sun using a solar position algorithm and calculates the desired angles of the heliostat. Afterwards, the

MCU uses two closed-loop controllers to send a PWM signal to each motor driver circuit. Two rotary encoders connected to the axes of the heliostat return the actual angular position of the heliostat to the MCU in order to close the control loop. The block diagram of the orientation control is shown in Figure 9.

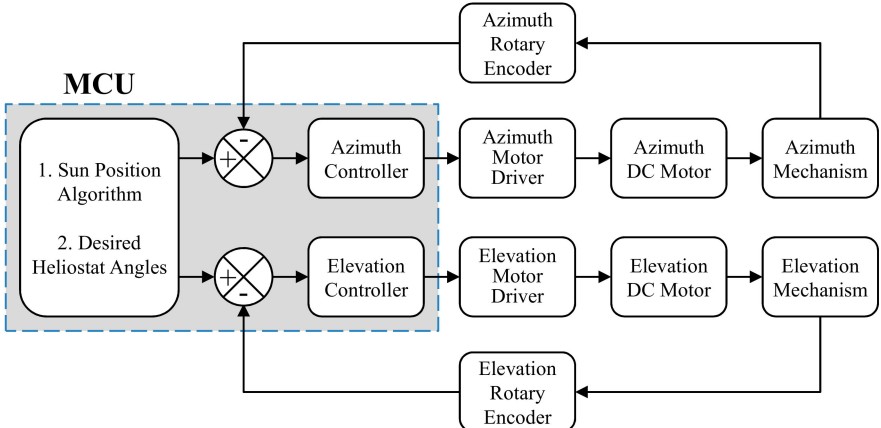

**Figure 9.** Block diagram of the orientation control algorithm.

### 2.6.1. Solar Position and Heliostat Angles

To guide the heliostat to the desired position, it is necessary to obtain the sun position at any time during the day. A reference system of horizontal coordinates describes the position of the sun in the sky by using the azimuth ($A_s$) and elevation angles ($E_s$). The azimuth angle is measured in regards to the South (positive to the West and negative to the East), whereas the elevation angle ranges from the horizon to the zenith, as shown in Figure 10. The azimuth and elevation angles of the sun position are given by a solar position algorithm with a maximal error of 0.0027° [21], which is sufficient for high concentration thermal systems, with a lower computational cost than fast algorithms. The algorithm takes the current date and time of the day and the geographical coordinates of the heliostat as input data. The algorithm also uses the local weather record to calculate the atmospheric refraction correction of the elevation angle of the sun, using the annual average local pressure and temperature.

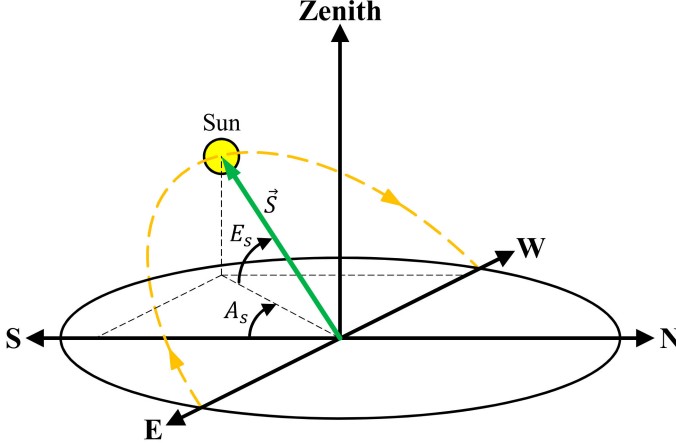

**Figure 10.** Solar vector.

In order to reflect the solar radiation on the desired target, the azimuth and elevation angles of the heliostat ($A_h$ and $E_h$) are given by the angles of the bisectrix of the angle formed by the solar vector ($\vec{S}$) and the fixed vector between the heliostat and the target ($\vec{T}$) [1], as shown in Figure 11. The bisector

between $\vec{S}$ and $\vec{T}$ must be equal to the normal vector of the heliostat reflecting surface $(\vec{N})$, which is given by the addition of the unit vectors of $\vec{S}$ and $\vec{T}$ as follows:

$$\vec{N} = \begin{bmatrix} \hat{S}_x + \hat{T}_x & \hat{S}_y + \hat{T}_y & \hat{S}_z + \hat{T}_z \end{bmatrix} \tag{1}$$

where $\hat{S}$ and $\hat{T}$ are the unit vectors of $\vec{S}$ and $\vec{T}$, and are given by Equations (2) and (3).

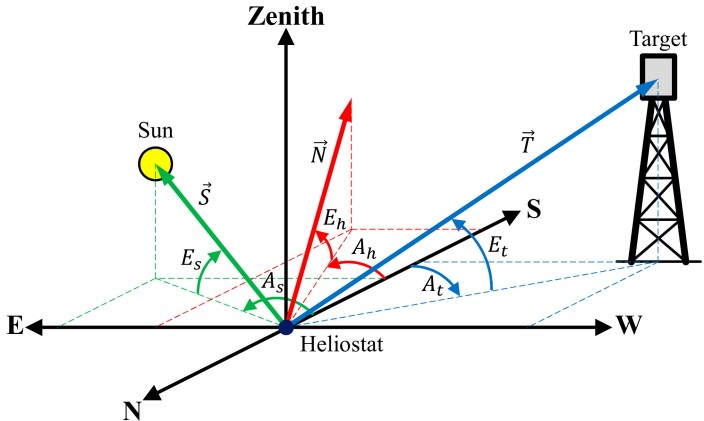

**Figure 11.** Heliostat angles.

$$\hat{S} = \begin{bmatrix} \sin(A_s)\cos(E_s) & \cos(A_s)\cos(E_s) & \sin(E_s) \end{bmatrix} \tag{2}$$

$$\hat{T} = \begin{bmatrix} \sin(A_t)\cos(E_t) & \cos(A_t)\cos(E_t) & \sin(E_t) \end{bmatrix} \tag{3}$$

Finally, the azimuth and elevation angles of the heliostat are given by Equations (4) and (5), respectively.

$$A_h = \tan^{-1}\left(\frac{\vec{N}_y}{\vec{N}_x}\right) \tag{4}$$

$$E_h = \tan^{-1}\left(\frac{\vec{N}_z}{\sqrt{\vec{N}_x^2 + \vec{N}_y^2}}\right) \tag{5}$$

### 2.6.2. Fuzzy Logic Controller

An FLC is applied for the position control of the DC motor of each axis. The controller takes the error signal ($e$) of the angular position to produce a control signal ($u$). This control technique emulates the experience of a human operator and has the advantage that it can deal with nonlinear systems and it is not necessary to know the mathematical model of the system. The FLC is graphically presented in Figure 12. A fuzzification block converts the numerical values of the controller inputs into linguistic variables (fuzzy sets). Later, an inference engine determines the value of the output fuzzy sets by using an "if-then" rule base. Finally, a defuzzification block combines the values of the scaled output fuzzy sets to get a crisp output value [22].

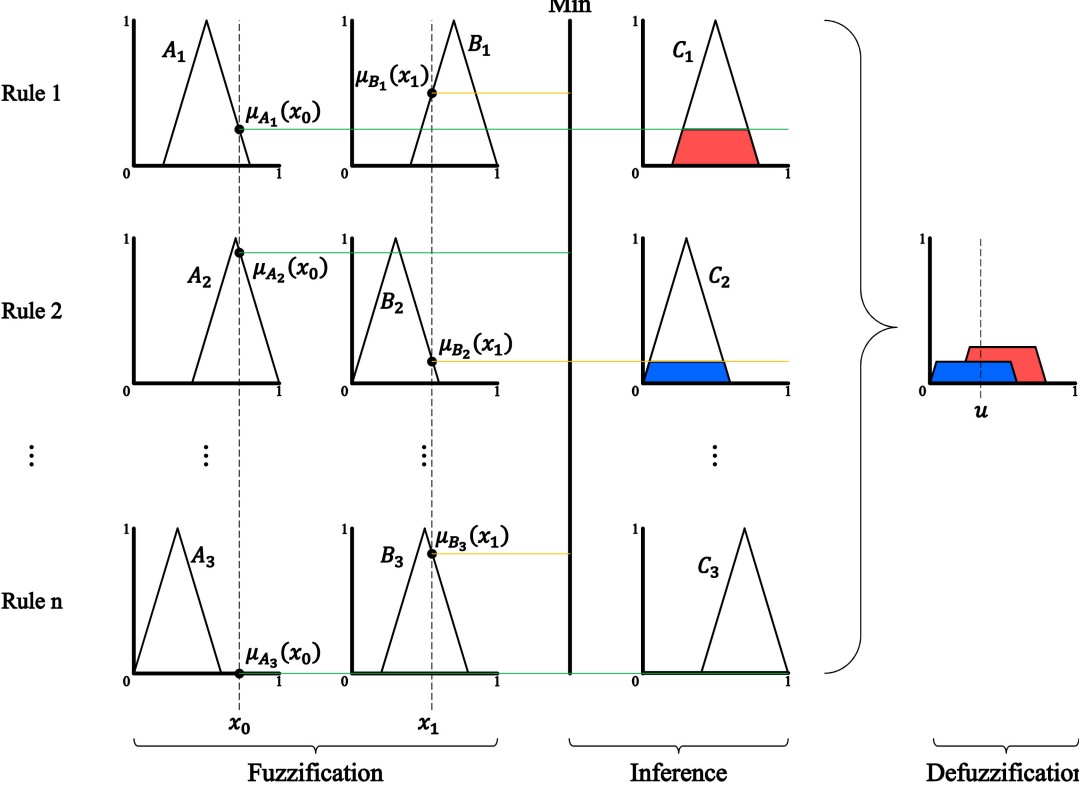

**Figure 12.** Fuzzy logic controller.

The block diagram of the proposed FLC is shown in Figure 13. It is a two-input (error *e* and the change of error *de*) and one-output (control signal *u*) controller. Furthermore, the FLC uses a nine rules rule base and the center of sums (CoS) defuzzification method, which is faster than many defuzzification methods due to its computational simplicity [23]. The rule base is shown in Table 4, whereas the CoS defuzzification method is given by Equation (6).

$$u = \frac{\sum_{i=1}^{n} [\mu(\overline{x}_i) A_i]}{\sum_{i=1}^{n} [A_i]} \tag{6}$$

where *u* represents the output value, *n* represents the number of the output fuzzy sets, $\overline{x}_i$ represents the centroid of the fuzzy set, and *A* represents the area of the scaled output fuzzy set *i*.

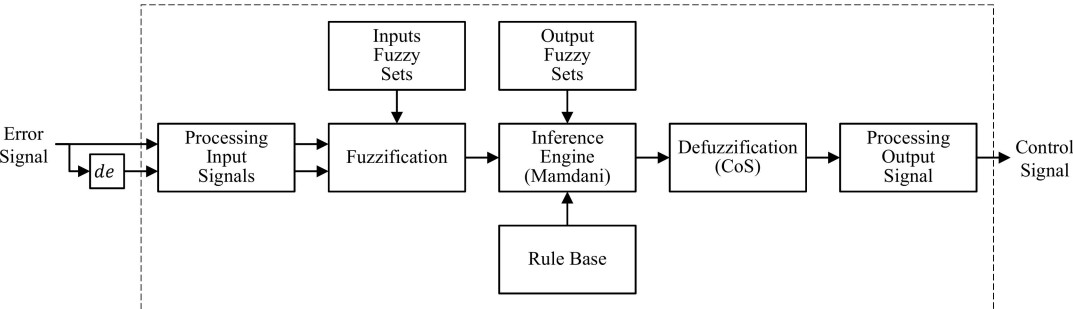

**Figure 13.** Block diagram of the FLC.

**Table 4.** Rule base of the fuzzy logic controller.

|  |  | *e* | | |
|---|---|---|---|---|
|  |  | **N** | **Z** | **P** |
|  | **N** | N | N | Z |
| *de* | **Z** | N | Z | P |
|  | **P** | Z | P | P |

Additionally, there is a conversion block in the inputs and output signal due to the difference between the signal values and the fuzzy sets values. The processing signals are given by:

$$e^* = \frac{e}{\pi} \tag{7}$$

$$de^* = \frac{(de)(T_s)}{\pi} \tag{8}$$

$$u^* = u V_{\max} \tag{9}$$

where $e^*$, $de^*$, and $u^*$ represent the scaled values of the inputs and the output, $T_s$ represents the sample time of the controller, and $V_{\max}$ represents the maximum DC motor supply voltage signal.

The fuzzy sets and the output surface of the FLC are shown in Figures 14 and 15, respectively. The fuzzy sets in each input signal and the output signal are denoted by N for the negative values, Z for the middle values and P for the positive values. The values of the membership functions were selected in order to the control signal of the FLC can modify the position of the DC motor due to the smallest change in the error signal.

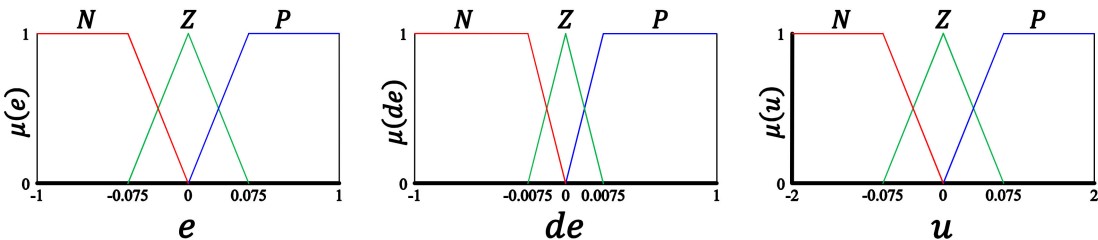

**Figure 14.** Fuzzy sets of the fuzzy logic controller.

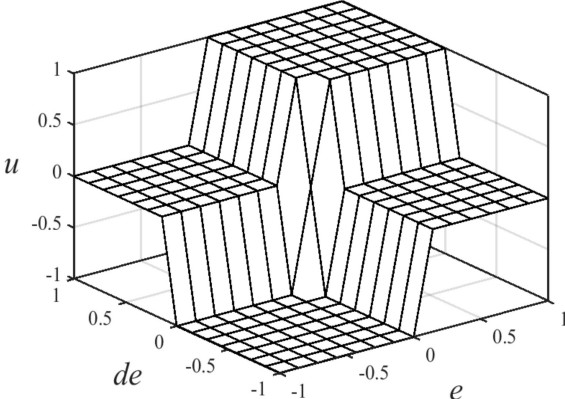

**Figure 15.** Output fuzzy surface.

2.6.3. Setpoint

The setpoint of the controller is established at a fixed period because it is not necessary to change the angular position of the heliostat every second of the day. The reference values of the azimuth and elevation angles are discretized every minute when the sun is above the horizon ($E_s > 0$), as shown in

Figure 16, taken on 9 May 2018. Both the desired angle and the reference value are the same when the value of the seconds is equal to 30.

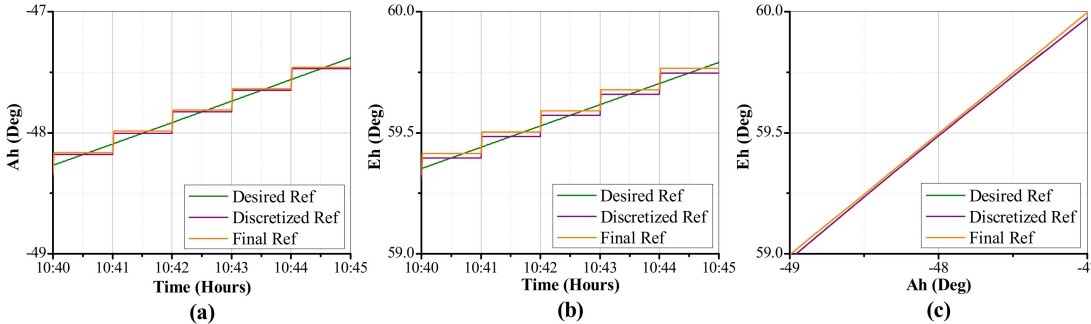

**Figure 16.** Setpoint of the orientation control for the azimuth angle (**a**), the elevation angle (**b**), and both angles (**c**).

To reduce the error signal of the discretized reference due to the resolution of the rotary encoders (0.0879° per encoder step), a final reference value is obtained converting the discrete reference value from radians to encoder steps and rounding it to the closest integer value, in order to the reference value in radians corresponds to a value in encoder steps, making the error signal in the controller to be zero when the heliostat reaches the desired position. The error values between the desired reference and the final reference are shown in Figure 17, taken on 9 May 2018.

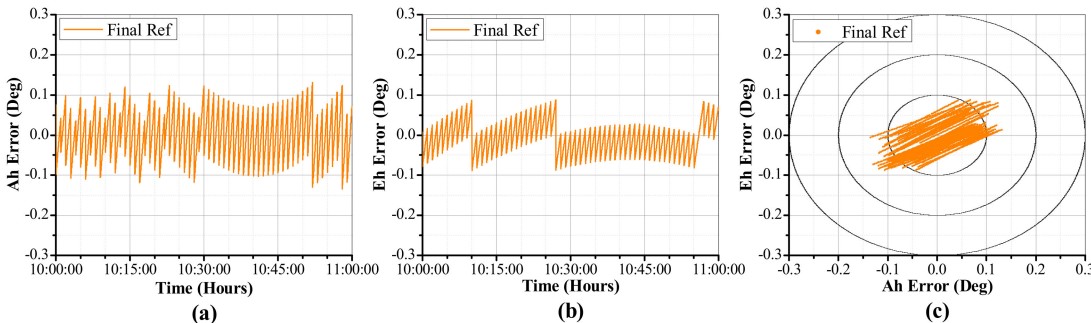

**Figure 17.** Final reference error for the azimuth angle (**a**), the elevation angle (**b**), and both angles (**c**).

## 3. Results and Discussion

The heliostat and the RTU of the control system are shown in Figures 18 and 19. The RTU consists of the PCB (Printed Circuit Board) of the MCU and the PCBs of the motor driver circuits. It is powered by a 24 V power supply for the motor driver circuits, and it is regulated at 5 V and 3.3 V in order to provide the power to the RTU components.

The developed GUI of the SU and the mobile App are shown in Figures 20 and 21. The GUI displays the values of time and date (date, local time, universal time, solar time, Julian day, day of the year, daylight saving time, and time zone), solar position (latitude and longitude, solar angles, declination, hour angle, right ascension, equation of time, dawn hour, sunset hour, solar noon, and daytime), heliostat data (latitude and longitude, heliostat angles, operation status, MAC address, an time and date), and the status of the communication with the RTU. There are also the control buttons and an image that shows the OS of the RTU. The mobile app consists of the buttons for the Bluetooth connection, labels that show the OS of the RTU, and the control buttons. The date and hour are colored in red if there is an error of one second in relation to the time of the device. Otherwise, the values are displayed in green. For both, the GUI and the mobile app, the control buttons are not activated until the communication with the RTU is established.

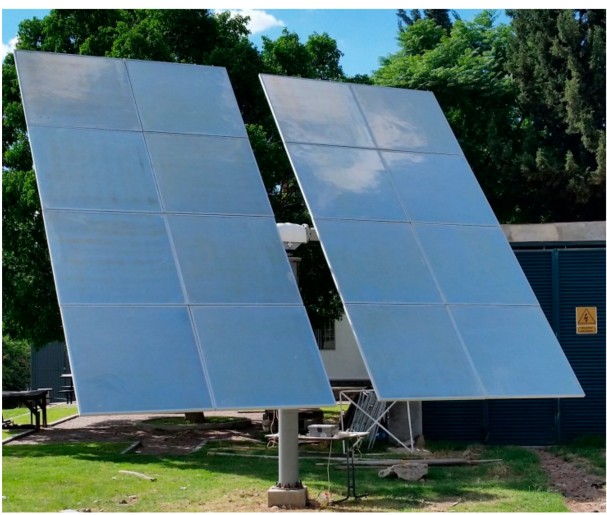

**Figure 18.** Heliostat.

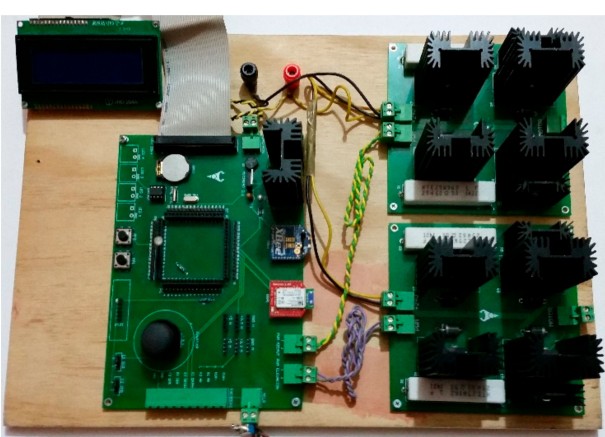

**Figure 19.** Remote terminal unit.

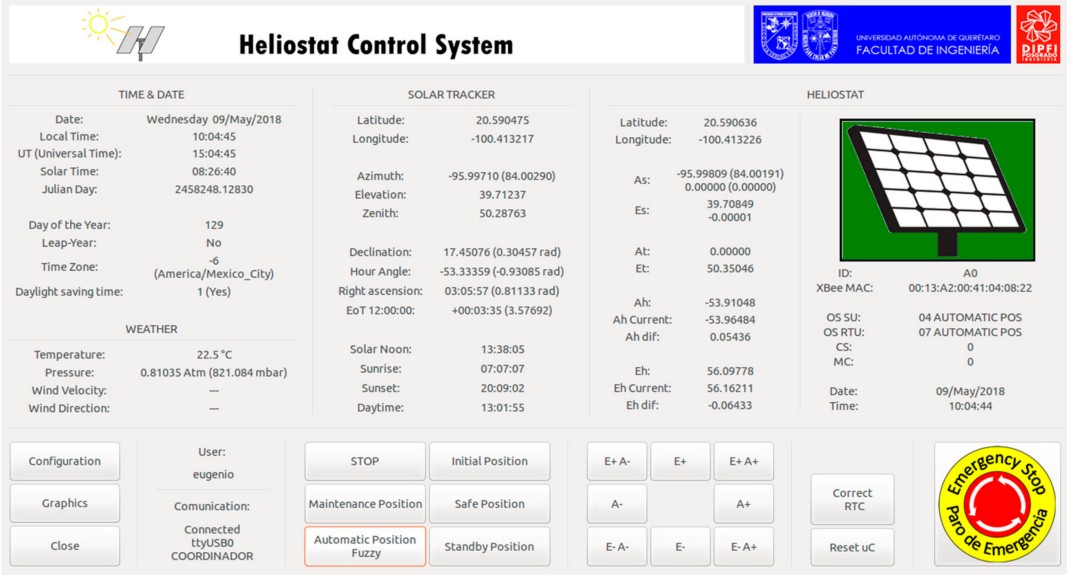

**Figure 20.** Graphical user interface.

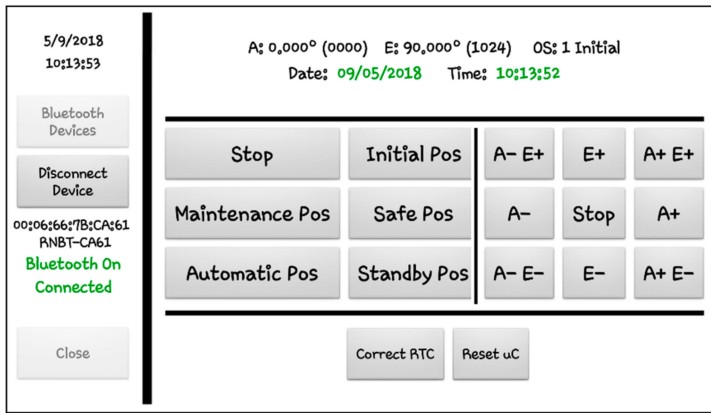

**Figure 21.** Mobile App.

The HCS was tested with the parameters shown in Table 5, first on the DC motors with no load and then on the heliostat. The experimental results of the orientation control are shown in Figure 22 for the DC motors with no load and in Figure 23 for the heliostat, whereas the values of the error of the orientation control are shown in Figure 24. Table 6 shows the mean squared error (MSE) of the orientation control compared to the three reference values.

**Table 5.** Parameters of the orientation control test.

| Parameter | Value |
|---|---|
| Date | 9 May 2018 |
| Time | 10:00:00–11:00:00 h |
| Latitude | 20.590636° N |
| Longitude | 100.413226° W |
| Time Zone | UTC-6 |
| Daylight saving time | Yes |
| Average Pressure | 821.084 mbar |
| Average Temperature | 22.5 °C |
| Average Wind speed | 6 m/s |
| Heliostat height | 2.85 m |
| Target height | 30 m |
| East-West distance to the target | 0 m |
| North-South distance to the target | 22.5 m North |

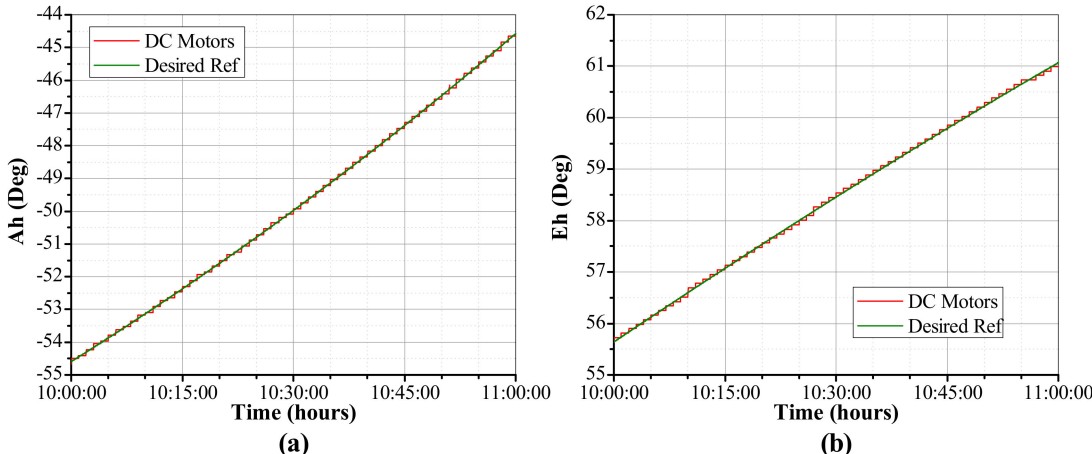

**Figure 22.** Orientation control response of the DC motors at no load for the azimuth angle (**a**) and the elevation angle (**b**).

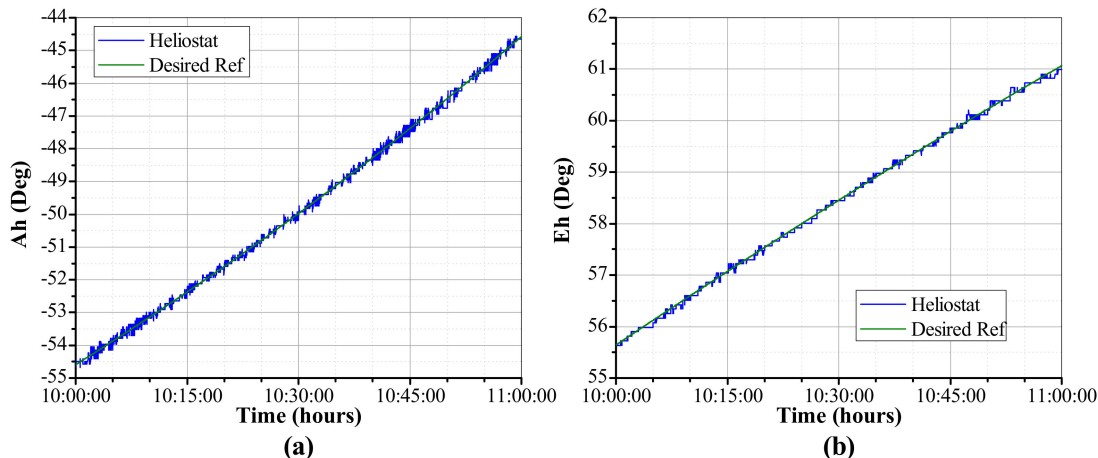

**Figure 23.** Orientation control response of the heliostat for the azimuth angle (**a**) and the elevation angle (**b**).

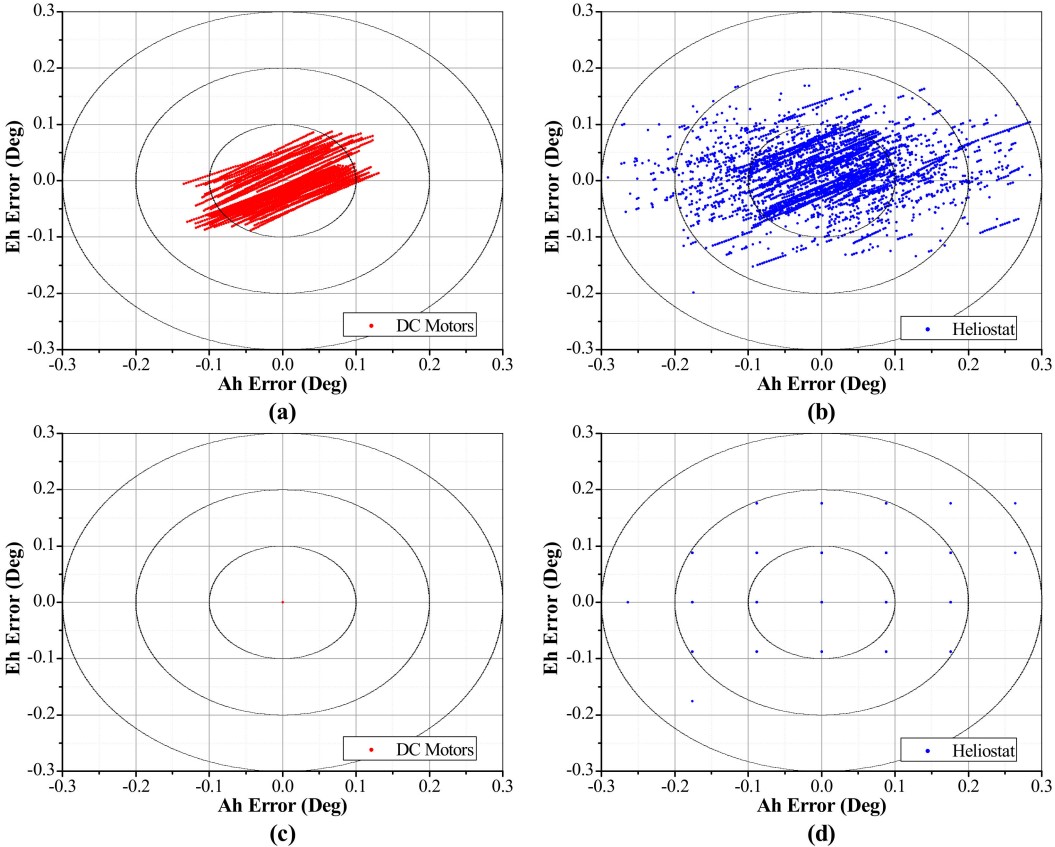

**Figure 24.** Error of the orientation control: for the DC motors with no load compared to the desired reference (**a**), for the heliostat compared to the desired reference (**b**), for the DC motors with no load compared to the final reference (**c**), and for the heliostat compared to the final reference (**d**).

**Table 6.** Orientation control mean squared error (MSE).

| Setpoint | DC Motors | | Heliostat | |
| --- | --- | --- | --- | --- |
| | **Azimuth** | **Elevation** | **Azimuth** | **Elevation** |
| Desired Ref | 0.05371° | 0.03761° | 0.09577° | 0.05401° |
| Discretized Ref | 0.02352° | 0.02730° | 0.08484° | 0.05568° |
| Final Ref | 0.0° | 0.0° | 0.08236° | 0.05511° |

The results show an MSE of 0.05371° and 0.03761° between the DC motors with no load and the desired reference for the azimuth and elevation angles, respectively (Figure 24a). Besides, there is an MSE of 0.0° between the final reference values and the angular position of the DC motors with no load (Figure 24c), and an MSE of 0.02352° and 0.02730° between the position of the DC motors with no load and the discretized reference for the azimuth and elevation angles, respectively. Therefore, the error signal in the controller is equal to zero when the angular position reaches the final reference value. For the orientation control of the heliostat, the results show an MSE of 0.09577° and 0.05401° for the azimuth and elevation angles compared to the desired reference (Figure 24b), which is less than 0.1° for both axes facing an average wind speed of 6 m/s over the mechanical structure. Finally, there is an MSE of 0.08236° for the azimuth axis and 0.05511° for the elevation axis compared to the final reference values (Figure 24d), with a maximum error value of 0.26367° (3 encoder steps).

## 4. Conclusions

A heliostat control system based on a SCADA system was designed and implemented in order to monitor and control the orientation of an Azimuth-Elevation heliostat. The proposed system allows a single registered operator to visualize and control the operation status of the heliostat in order to detect any failure in its operation. The system consists of a supervisory unit with a graphical user interface, a remote terminal unit implemented in a low-cost microcontroller, a wireless communication network, and a mobile app. Besides, a fuzzy logic controller embedded in the remote terminal unit orients the heliostat to the desired position.

The stand-alone RTU performs all the calculations and tasks of the orientation control by only a control command sent by the supervisory unit. Therefore, this reduces the network traffic between the SU and the RTU. The experimental results show a similar behavior between the orientation control of the DC motors with no load and the heliostat, with an MSE less than 0.1° for the orientation control of the heliostat, despite the backlash in the axis mechanisms and the load of the wind over the mechanical structure. Therefore, the control algorithm with fuzzy logic can guide the heliostat to the desired angles using the same parameters of the position control of the DC motors with no load, and it is not necessary to tune the controller parameters if the dynamics of the system changes, as in the traditional PID controllers.

As future work, the system can be applied to control an entire heliostat field just by modifying the GUI in order to send the corresponding control command to each heliostat by using the broadcast model of the RF modules, and the same program code for each RTU, only by updating the data of the geographical position and the distance to the target of each heliostat. Besides, the FLC can orient all the heliostats by using the same control parameters, because it is robust in front of changes in the dynamics of the process. However, higher resolution encoders would be necessary for farthest heliostats from the central tower. The system can also be adapted in order to control other two-axis solar-tracking systems that only use the solar-tracking system, such as photovoltaic, parabolic trough, or solar dish systems.

**Author Contributions:** Conceptualization, E.S.-P. and M.T.-A.; methodology, E.S.-P., E.A.R.-A., and R.V.C.-S; software and hardware, E.S.-P.; validation, E.S.-P. and R.V.C.-S.; project administration, M.T.-A.

**Funding:** This research received no external funding.

**Acknowledgments:** The authors would like to thank Consejo Nacional de Ciencia y Tecnología (CONACYT-México) for supporting this research.

**Conflicts of Interest:** The authors declare no conflict of interest.

## Abbreviations

| | |
|---|---|
| API | Application Programming Interface |
| CoS | Center of Sums |
| FLC | Fuzzy Logic Controller |
| GUI | Graphical User Interface |
| HCS | Heliostat Control System |
| I2C | Inter-Integrated Circuit |
| ICSP | In-Circuit Serial Programming |
| LCD | Liquid Crystal Display |
| LDR | Light Dependent Resistors |
| MCU | Microcontroller Unit |
| MDS | Microprocessor Driver Systems |
| MSE | Mean Squared Error |
| OS | Operation Status |
| PID | Proportional–Integral–Derivative |
| PV | Photovoltaic |
| PWM | Pulse Width Modulation |
| RF | Radio Frequency |
| RTC | Real-Time Clock |
| RTU | Remote Terminal Unit |
| SCADA | Supervisory Control And Data Acquisition |
| SDS | Sensor Driver Systems |
| SSI | Synchronous Serial Interface |
| SU | Supervisory Unit |
| UART | Universal Asynchronous Receiver-Transmitter |

## Symbols

| | |
|---|---|
| $\vec{S}$ | Solar vector |
| $\vec{T}$ | Target vector |
| $\vec{N}$ | Normal vector of the heliostat |
| $\hat{S}$ | Solar unit vector |
| $\hat{T}$ | Target unit vector |
| $A_s$ | Solar vector azimuth angle |
| $E_s$ | Solar vector elevation angle |
| $A_t$ | Target vector azimuth angle |
| $E_t$ | Target vector elevation angle |
| $A_h$ | Heliostat azimuth angle |
| $E_h$ | Heliostat elevation angle |
| $e$ | Controller error signal |
| $de$ | Controller change of error signal |
| $u$ | Controller output signal |
| $T_s$ | Controller sampling time |
| $V_{max}$ | Controller maximum output voltage |

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
