# Peer review of "SCADA-Based Heliostat Control System with a Fuzzy Logic Controller for the Heliostat Orientation"

_applsci, doi:10.3390/app9152966_

Round 1
Reviewer 1 Report
This paper could be of interest to the readers of Applied Sciences.
In general, it has a good organization and is well presented.
However, it needs some specific revision.
2.2. Mechanical structure
Please, could you provide the characteristics of the mechanical structure of the heliostat
2.6 PWM motor driver
Further specific characteristics of the motors driver circuit are welcomed.
Figure 6 needs to be described more in detail.
e.g. How much is the request power?
.
2.6.4 Setpoint
How much is the precision of the encoder?
In other words how much could be the difference between the desired ref and the final ref?
Reviewer 2 Report
By a scientific point of view, the research work appear accurately studied and correctly experimented. The work is of relevance specifically for heliostats plants but more generally, these tracking techniques could be applied in other systems for solar energy exploitation.
The level of English is good and the scientific terminology appropriate. In contrast, punctuation should be checked (e.g. delete the full stop at the end of line 72, insert a comma before each “which”).
Maybe it would be useful for the readers to explain the terms “SCADA” and “PID” cited in the abstract (the nomenclature table is only at the end of the article).
The discussion of the state of the art is quite succinct, even though the examined bibliography is large and significant. The illustration of each of the cited articles is extremely synthetic. Perhaps the most significant works could be described with some additional details, especially by a practical point of view.
System components and methodology are briefly explained in short dedicated subsections. The synthesis of the presentation is appreciable, moreover the splitting of the arguments defines the single elements and clarifies the development of the work. The practical application and the characteristics of the electronic components used are discussed with experimental details and reporting the values of the use parameters.
The system has been presented with few details about location and environmental conditions of the actual site (Table 3 lists the key parameters of the site and some dimensions of the solar plant).
This highlights the broad application possibilities of the proposed system. Nevertheless, the solar plant description could be enriched with some extra information regarding the actual facility utilized for the experimentation.
The Figures are informative and most of them are well presented. Only for some of them (Fig. 16, Fig. 19, Fig. 20; perhaps also Fig. 8 and Fig. 21) the readability is difficult: it would be advisable to use larger fonts and to present them in a bigger size.
Tables visually collect features, parameters, data and results. However, the validation of numerical results, and more generally the validation of the tracking methodology, should be better explained and presented in a more extensive way.
The experiment shown concerns only one heliostat. The positive results are encouraging, but the real challenge will be the application in a multi-heliostat field.
In an actual field of heliostats, the management of hundreds of mirrors (with crossed interactions and environmental effects) multiplies the difficulties.
In consideration of this, please complete the conclusion with advantages and disadvantages of the proposed system and of the practical application of the specific system components.
